

# An enhanced Genetic Folding algorithm for prostate and breast cancer detection

Mohammad A. Mezher[1], Almothana Altamimi[2] and Ruhaifa Altamimi[3]

[1] College of Computing, Fahad Bin Sultan University, Tabuk, Saudi Arabia

[2] Department of Clinical Medicine and Surgery, Università degli Studi di Napoli Federico, di Napoli Federico, Italy

[3] Department of Business and Data Analytics, University of Huddersfield, Huddersfield, United Kingdom

Corresponding author
Mohammad A. Mezher,
mmezher@fbsu.edu.sa

## ABSTRACT

Cancer's genomic complexity is gradually increasing as we learn more about it. Genomic classification of various cancers is crucial in providing oncologists with vital information for targeted therapy. Thus, it becomes more pertinent to address issues of patient genomic classification. Prostate cancer is a cancer subtype that exhibits extreme heterogeneity. Prostate cancer contributes to 7.3% of new cancer cases worldwide, with a high prevalence in males. Breast cancer is the most common type of cancer in women and the second most significant cause of death from cancer in women. Breast cancer is caused by abnormal cell growth in the breast tissue, generally referred to as a tumour. Tumours are not synonymous with cancer; they can be benign (noncancerous), pre-malignant (pre-cancerous), or malignant (cancerous). Fine-needle aspiration (FNA) tests are used to biopsy the breast to diagnose breast cancer. Artificial Intelligence (AI) and machine learning (ML) models are used to diagnose with varying accuracy. In light of this, we used the Genetic Folding (GF) algorithm to predict prostate cancer status in a given dataset. An accuracy of 96% was obtained, thus being the current highest accuracy in prostate cancer diagnosis. The model was also used in breast cancer classification with a proposed pipeline that used exploratory data analysis (EDA), label encoding, feature standardization, feature decomposition, log transformation, detect and remove the outliers with Z-score, and the BAGGINGSVM approach attained a 95.96% accuracy. The accuracy of this model was then assessed using the rate of change of PSA, age, BMI, and filtration by race. We discovered that integrating the rate of change of PSA and age in our model raised the model's area under the curve (AUC) by 6.8%, whereas BMI and race had no effect. As for breast cancer classification, no features were removed.

## INTRODUCTION

Cancer is, at times, a severe disease or set of diseases that have historically been the most prevalent and challenging to treat (*Adjiri, 2016*). It is commonly defined as the abnormal proliferation of various human cells, thus resulting in its etiological heterogeneity (*Cooper, 2000*). This abnormal growth can be classified into two subsets: (a) malignant and (b) benign. Benign tumours stay localized to their original site, whereas malignancies can

invade and spread throughout the body (metastasize) (*Cooper, 2000*). Breast cancer recently eclipsed lung cancer as the most prominent cancer subtype worldwide, equalling 11.7% (2.3 million) of the new cancer cases (*Sung et al., 2021*).

Prostate cancer is one of the most prevalent male malignancies globally, contributing 7.3% of the estimated incidence in 2020 (*Sung et al., 2021*). This amounted to 1,414,259 new cases and 375,304 deaths from this disease (*Sung et al., 2021*). The prostate is a dense fibromuscular gland shaped like an upside-down cone. It is located around the neck of the urinary bladder, external to the urethral sphincter and functions in a supportive role in the male reproductive system. The alkaline fluid it secretes into semen protects sperm cells from the acidic vaginal environment (*Singh & Bolla, 2021*).

Although early prostate cancer is typically asymptomatic, it may manifest itself in the form of excessive urination, nocturia, haematuria, or dysuria (*Leslie et al., 2021*). Classically, prostate cancer is detected with a Digital Rectal Examination (DRE) and a blood test for prostate-specific antigen (PSA) (*Descotes, 2019*). TRUS-guided biopsy continues to be the gold standard for confirming diagnoses, despite its 15–46% false-negative rate and up to 38% tumour under grading rate compared to the Gleason score (*Descotes, 2019*; *Kvåle et al., 2009*).

However, prostate cancer development's aetiology and mechanisms are still determined (*Howard et al., 2019*). The different mechanisms that develop prostate cancer ultimately affect the therapy proposed. Hence, patient stratification and tumour classification are vital. Especially true considering that prostate cancer is heterogeneous in a clinical, spatial and morphological aspect (*Tolkach & Kristiansen, 2018*). The following prostate cancer stages of development are currently posited: intraepithelial-neoplasia, androgen-dependent adenocarcinoma, and androgen-independent or castration-resistant cancer (CRC) (*Howard et al., 2019*). Prostate cancer heterogeneity is further highlighted in a recent study shedding light on mRNA expressional variations from a normal prostate to full metastatic disease (*Marzec et al., 2021*). This means that metastatic CRC (mCRC) tumours are more complicated than primary prostate tumours. An association that is further aggravated when genomics come into play.

Prostate cancer development and progression have been heavily linked to Androgen Receptor (AR) signalling pathway; thus, androgen deprivation therapy (ADT) has been used for patients who have advanced prostate cancer (*Hatano & Nonomura, 2021*). Although primarily effective, a large proportion of patients develop androgen-independent or CRC. Thus, pharmacological therapies are considered, including abiraterone, enzalutamide, docetaxel and radium-223 (*Howard et al., 2019*).

As mentioned above, breast cancer has become the most common cancer diagnosed, eclipsing lung cancer. The etiology of breast cancer is multi-faceted; many risk factors play a role in the likelihood of a diagnosis; these risk factors can be sub-divided into seven groups. (1) age (2) gender (3) previous diagnosis of breast cancer (4) histology (5) family history of breast cancer, (6) reproduction-related risk factors and (7) exogenous hormone use (*Alkabban & Ferguson, 2022*). Race may also indicate a higher prevalence in non-Hispanic white individuals than African Americans, Hispanics, Native Americans, and Asian Americans (*Alkabban & Ferguson, 2022*).

Many modalities of screening exist with varying specificities and sensitivities to breast cancer. Mammography is one of the front-line tests to screen for breast cancer, with its sensitivity ranging from 75–90% and its specificity ranging from 90–95% (*Bhushan, Gonsalves & Menon, 2021*). Screening is done at a macro level to determine whether the growth is malignant or benign. However, classification at the micro-level is required to identify the molecular basis behind breast cancer progression to target specific therapies. Luminal-A tumours (58.5%) are the most prevalent subtype of breast cancer tumours, then triple-negative (16%), luminal-B (14%) and HER-2 positive (11.5%) being the least prevalent (*Al-thoubaity, 2019*).

Breast cancer tumours are classified using the TNM classification system in which the primary tumour is denoted as T, the regional lymph nodes as N, and distant metastases as M. Breast cancer can also be classified about its invasiveness, with lobular carcinoma *in situ* (LCIS) and ductal carcinoma *in situ* (DCIS) being non-invasive. Invasive ductal cancer accounts for 50–70% of invasive cancers, while invasive ductal cancer accounts for 10% (*Alkabban & Ferguson, 2022*).

Once the molecular basis of breast cancer is identified, treatment can begin. Many chemotherapeutic agents are used in the treatment of breast cancer, including Tamoxifen (ER-positive breast cancer), Pertuzumab (HER2-overexpressing breast cancer) and Voxtalisib (HR+/HER- advanced breast cancer) (*Bhushan, Gonsalves & Menon, 2021*).

This article aims to develop a proper kernel function using GF to separate benign prostate and breast cells from tumour-risk genetic cells using SVM, compared to six machine learning algorithms. In addition, the proposed GF-SVM classifier can classify breast and prostate cancer cells better than the six different classifiers. We have also included all features found in the datasets to test the ability of the proposed modelling to predict the best accuracy. The proposed GF-SVM implementation is reliable, scalable, and portable and can be an effective alternative to other existing evolutionary algorithms (*Mezher, 2022*).

## Literature review

Artificial intelligence (AI) is revolutionizing healthcare, incredibly patient stratification and tumour classification, which are pivotal components of targeted oncology. Training a machine learning (ML) model to analyze large datasets means more accurate prostate cancer diagnoses (*Tătaru et al., 2021*). Artificial neural networks (ANN) is a tool that has been used in advanced prognostic models for prostate cancer (*Jović et al., 2017*). Other models employed in the classification of cancers based on gene expression include K-Nearest Neighbor (KNN) and support vector machines (SVM). The accuracy of these models varies between 70% and 95%, depending on the number of genes analyzed. Previous studies have used these and other ML models to classify prostate cancer (*Bouazza et al., 2015*); thus, we aim to use the Genetic Folding (GF) algorithm in classifying patients with prostate cancer. *Alba et al. (2007)* used a hybrid technique for gene selection and classification. They presented data of high dimensional DNA Microarray. They found it initiates a set of suitable solutions early in their development phase. Similarly, *Tahir & Bouridane (2006)* found that using a hybrid algorithm significantly improved their results

in classification accuracy. However, they concluded their algorithm was generic and fit for diagnosing other diseases such as lung or breast cancer. *Bouatmane et al. (2011)* used an RR-SFS method to find a 99.9% accuracy. This was alongside other classification methods such as bagging/boosting with a decision tree. *Lorenz et al. (1997)* compared the results from two commonly used classifiers, KNN and the Bayes classifier. They found 78% and 79%, respectively. They had the opportunity to improve these results for ultrasonic tissue characterisation significantly.

Developing a quantitative CADx system that can detect and stratify the extent of breast cancer (BC) histopathology images was the primary clinical objective for *Basavanhally et al. (2010)*, as they demonstrated in their article the ability to detect the extent of BC using architectural features automatically. BC samples with a progressive increase in Lymphocytic Infiltration (LI) were arranged in a continuum. The region-growing algorithm and subsequent MRF-based refinement allow LI to isolate from the surrounding BC nuclei, stroma, and baseline level of lymphocytes (*Basavanhally et al., 2010*). The ability of this image analysis to classify the extent of LI into low, medium and high categories can show promising translation into prognostic testing.

A meta-analysis identified several trends concerning the different types of machine learning methods in predicting cancer susceptibility and outcomes. It was found that a growing number of machine learning methods usually improve the performance or prediction accuracy of the prognosis, particularly when compared to conventional statistical or expert-based systems (*Cruz & Wishart, 2006*). There is no doubt that improvements in experimental design alongside biological validation would enhance many machine-based classifiers' overall quality and reproducibility (*Cruz & Wishart, 2006*). As the quality of studies into machine learning classifiers improves, there is no doubt that they will become more routine use in clinics and hospitals.

## The proposed model
### Dataset

The prostate dataset of 100 patients (see Table 1) was used to test the proposed GF algorithm and analyze the outcomes. The dataset consists of 100 observations and nine variables; Radius (Rad.), Texture (Text.), Perimeter (Perim.), Area, Smoothness (Smooth), Compactness (Compact), Symmetry, Fractal_dimension (Fractal_dim.), and Diagnosis.

The breast cancer dataset is a repository maintained by the University of California. The dataset contains 569 samples of malignant and benign tumor cells. Columns 1–30 contain real-value features that have been computed from digitized images of the cell nuclei. Table 2 shows the first seven columns found in the breast dataset (*University of California Irvine, 2019*).

All the prostate cancer dataset observations included in this experiment have been collected from Kaggle.com (*Saifi, 2018*). Prostate and breast cancer patients were labeled with M, whereas those without the cancer were labeled with B, as seen in the following tables.

**Table 1  Samples of prostate cancer dataset.**

| Rad. | Text. | Perim. | Area | Smooth | Compact | Symmetry | Fractal_dim. | Diagnosis |
|------|-------|--------|------|--------|---------|----------|--------------|-----------|
| 23 | 12 | 151 | 954 | 0.143 | 0.278 | 0.242 | 0.079 | M |
| 9 | 13 | 133 | 1326 | 0.143 | 0.079 | 0.181 | 0.057 | B |
| 21 | 27 | 130 | 1203 | 0.125 | 0.16 | 0.207 | 0.06 | M |
| 14 | 16 | 78 | 386 | 0.07 | 0.284 | 0.26 | 0.097 | M |
| 9 | 19 | 135 | 1297 | 0.141 | 0.133 | 0.181 | 0.059 | M |
| 25 | 25 | 83 | 477 | 0.128 | 0.17 | 0.209 | 0.076 | B |
| 16 | 26 | 120 | 1040 | 0.095 | 0.109 | 0.179 | 0.057 | M |
| 15 | 18 | 90 | 578 | 0.119 | 0.165 | 0.22 | 0.075 | M |
| 19 | 24 | 88 | 520 | 0.127 | 0.193 | 0.235 | 0.074 | M |
| 25 | 11 | 84 | 476 | 0.119 | 0.24 | 0.203 | 0.082 | M |

### Support vector machine

This section will go through the basic SVM concepts in the context of two-class classification problems, either linearly or non-linearly separable. SVM is based on the Vapnik–Chervonenkis (VC) theory and the Structural Risk Minimization (SRM) principle (*Shawe-Taylor et al., 1996*). The goal is to identify the optimum trade-off between lowering training set error and increasing the margin to get the highest generalization ability while resisting overfitting. Another significant benefit of SVM is convex quadratic programming, which generates only global minima, preventing the algorithm from being trapped in local minima. *Cristianini & Shawe-Taylor (2000)* provide in-depth explanations of SVM theory. The remaining part of this section will go through the fundamental SVM ideas and apply them to classic linear and nonlinear binary classification tasks.

*Linear margin kernel classifier*  Suppose a binary classification problem is given as $\{x_i, y_i\}$, $x_i \in \Re N$ and a set of corresponding labels are notated $y_i \in \{-1,1\}$ for $i = 1, 2, \ldots, N$, where $\Re N$ denotes vectors in a d-dimensional feature space. A hyperplane provided in SVM separates the data points using Eq. (1):

$$(x_i) = w^T \cdot X + b = 0 \tag{1}$$

where **w** is an n-dimensional coefficient vector that is normal to the hyperplane and $b$ is the offset from the origin and X is the features in a sample.

*Nonlinear Margin Kernel Classifier*  It has been shown that the input space can be mapped onto a higher-dimensional feature space where the training set cannot be separated. The input vectors are mapped into a new, higher-dimensional feature space, indicated as $F : \Re N \to H^K$, where N <K, to produce the SVM model:

$$F(x) = w^T \cdot K(x_i, x_j) + b. \tag{2}$$

The mapping's functional $(x)$, is implicitly determined by the kernel trick $K(x_i, x_j) = \varphi(x_i) \cdot \varphi(x_j)$, which is the dot product of two feature vectors in extended feature space. In SVMs, an extensive range of kernel tricks are accessible for application. The most common SVM kernels are shown in Table 3. In the table, $\gamma$ and $p$ are predefined user parameters.

**Table 2 Samples of breast cancer dataset.**

| Radius | Texture | Perimeter | Area | Smoothness | Compactness | Concavity |
|---|---|---|---|---|---|---|
| 20.57 | 17.77 | 132.90 | 1326.0 | 0.08 | 0.08 | 0.09 |
| 19.69 | 21.25 | 130.00 | 1203.0 | 0.11 | 0.16 | 0.20 |
| 20.29 | 14.34 | 135.10 | 1297.0 | 0.10 | 0.13 | 0.20 |
| 12.45 | 15.70 | 82.57 | 477.10 | 0.13 | 0.17 | 0.16 |
| 18.25 | 19.98 | 119.60 | 1040.0 | 0.09 | 0.11 | 0.11 |
| 13.71 | 20.83 | 90.20 | 577.90 | 0.12 | 0.16 | 0.09 |

**Table 3 The traditional Kernel functions.**

| Name | Kernel function |
|---|---|
| Linear Kernel | $k\left(x_i, x_j\right) = x_i \cdot x_j$ |
| Polynomial | $k\left(x_i, x_j\right) = (x_i \cdot x_j + 1)^p$ |
| RBF | $k\left(x_i, x_j\right) = exp^{-\gamma(x_i - x_j)^2}$ |

### Genetic folding algorithm

The GF algorithm was created by *Mezher & Abbod (2011)*. The GF algorithm's fundamental principle is to group math equations using floating numbers. Floating numbers are created by taking random samples of operands and operators. In GF, the linear chromosomes serve as the genotype, while the parse trees serve as the phenotype. This genotype/phenotype technique works well for encoding a lengthy parse tree in each chromosome. The GF approach has shown to be effective in various computer problems, including binary, multi-classification, and regression datasets. For example, GF has been proven to outperform other members of the evolutionary algorithm family in binary classification, multi-classification (*Mezher & Abbod, 2014*), and regression (*Mezher & Abbod, 2012*).

On the other hand, GF may yield kernels employed in SVMs. Kernels are created by combining operands and operators to generate appropriate pairings, such as (3 + 4). At each pair, the pair is indexed at random in a GF array cell (known as GF kernel). Correlation between pairings is boosted by picking pairs at random that boost the strength of the generated GF kernels. Each GF kernel chromosome was divided into a head segment that only carries functions and a tail segment containing terminals. However, the size of the head segment must be determined ahead of time, but the size of the tails segment does not need to be determined since the GF algorithm predicts the number of genes needed based on the pairs required for the drawn functions (*Mezher & Abbod, 2017*). Furthermore, the GF algorithm predicts the number of operands (terminals) necessary each time the GF algorithm generates different operators (function) at random. The proposed modified GF algorithm has the pseudo-code that can be seen in Fig. 1:

The following operands and operators were used to forecast malignant and benign cells in the prostate dataset, as shown in Table 4:

The GF genome comprises a continuous, symbolic string or chromosome of equal length. Table 5 shows an example of a chromosome with varying pairs that may be used to create a valid GF kernel.

```
#Initialization
# Set the operators and operands needed along with the size of GF chromosome
input Op_k = {plus_v, plus_s, minus_v, minus_s, multi_s,}, Operand_n = {1,.., n-1},
Len = GF chromosome length
# Initialize the pool of GF pairs
set S = (indx_0, operand_1, Op_1, operand_2), …, (indx_{n-1}, operand_n, Op_k, operand_n)
# GF chromosome/population generation
Algorithm GFs
    1.  For pop= 1 to number_of_generation
    2.      For i=1 to len
    3.          M= Select (Op_i, Operands) from S
    4.          GF_chromosome = concatenate (GF_chromosome, M)
    5.      End for
    6.      Construct pop = pop + GF_chromosome
    7.      Fitness = SVM(GF_chromosome)
    8.      pop = pop + 1
    9.  End for
    10. New kernels are generated to the most fitness
    11. Result: Optimum GF kernel that have highest accuracy
```

**Figure 1  Pseudo-code of GF algorithm.**

**Table 4  The symbols used to formulate kernel functions.**

| Type of symbols | Name | No. of arity |
|---|---|---|
| **Operators** | 'Plus_s' | 1 |
| | 'Minus_s' | 1 |
| | 'Multi_s' | 1 |
| | 'Plus_v' | 2 |
| | 'Minus_v' | 2 |
| **Operands** | 'x' | 1 |
| | 'y' | 1 |

**Table 5  The encoding/decoding used to formulate kernel functions.**

| Index | 0 | 1 | 2 | 3 | 4 |
|---|---|---|---|---|---|
| **GF Decoding** | 'Minus_s', | 'Plus_s', | 'y' | 'y' | 'x' |
| **GF Encoding** | '1.2' | '3.4' | '0.2' | '0.3' | '0.4' |

## Model evaluations

The classification performance of each model is evaluated using statistical classification accuracy. Equation (3) is used to determine the accuracy of the GF algorithm's correctly

classified instances:

$$Accuracy = \frac{TP + TN}{TP + TN + FP + FN} * 100\%. \tag{3}$$

The True Positive (TP), True Negative (TN), False Positive (FP), and False Negative (FN) define this accuracy measurement. A TP is made when the algorithm predicts malignant (positive), and the actual result is malignant (positive). A TN is made when the algorithm predicts benign (negative), and the actual result is benign (negative). FP occurs when the algorithm predicts a benign (negative) instance as malignant (positive). Finally, when the GF algorithm classifies a malignant (positive) instance as benign (negative), the result is FN. The accuracy performance metrics are compared in Table 5.

We have also included another evaluating estimator to measure the quality of the proposed mode to choose the best. Some of the estimators tolerate the small samples better than others, while other measures the quality of the estimator with large samples. Mean Square Error (MSE) (*Schluchter, 2014*) has been used as a standard metric to measure model performance in medical, engineering, and educational studies. Assume E' is the predicted classes E' ={(pred(y1), pred(y2), pred(y3), …} of the observed classes E ={y1, y2, y3,..}, then the MSE is defined as the expectation of the squared deviation of E' from E:

$$MSE\left(E'\right) = \sum_{i=1}^{n} \frac{(E' - E)^2}{n}. \tag{4}$$

MSE measures the estimator's bias (accuracy), which shows how much its predicted value deviates consistently from the actual value and the estimator's variance (precision), which indicates how much its expected value fluctuates due to sampling variability. Squaring the errors magnifies the effect of more significant errors. These calculations disproportionately penalize larger errors more than more minor errors. This attribute is critical if we want the model to be as accurate as possible.

## RESULTS

We are preprocessing the dataset using min-max standardization, without removing any features that prevented incorrect relevance assignments. The elimination of the features showed either weakness or strength correlation and reduced the accuracy values, as shown in Table 5. In statistics, correlation is any connection between two variables but does not indicate causation, implying that the variables are dependent on one another. The closer a variable is to 1, the more significant the positive correlation; the closer a variable is to −1, the stronger the negative correlation; and the closer a variable is to 0, the weaker the negative correlation. Each malignant datapoint was subjected to analysis. We implemented the proposed method using Visual Studio Code by Python.

This article conducted our experiments on 100 patients of eight features collected from kaggle.com (*Saifi, 2018*) for prostate cancer and 596 patients of 30-features for the breast cancer dataset. The instances were classified as malignant or benign in the experiments. The following are the steps involved in conducting the GF algorithm to produce a valid GF kernel. GF starts by correctly generating the initial GF genes (operators, operands) (operand, operator, operand). Then, the GF algorithm generates a valid GF chromosome

containing 50 genes. Based on a five fold cross-validation approach, the dataset was divided into a training set and a testing set. The detailed GF algorithm is explained in Fig. 1.

Figures 2 and 3A–3E depict the complexity, diversity, ROC curve, accuracy, and best GF chromosome results for prostate and breast datasets. Figures 2 and 3A show the GF algorithm's performance concerning the complexity of the generic kernel. Figures 2 and 3B show the population's diversity in each generation, where GF maintains the best fitness values with a chance of accepting weak kernels. Figures 2 and 3C shows that GF generic kernel was the best Area Under Curve (AUC) value compared with conventional SVM classifiers; linear, RBF, and polynomial kernel functions. Based on the MSE, the proposed model shows (Figs. 2 and 3E) minor differences between the observed values and the predicted classes. In Fig. 2E, the proposed model beats the other kernels with at least the average of errors was better than the minimum box plot of the best model (RBF kernel). Figure 2E shows a perfect model with no error produced of almost zero variance. In order to show the performance of the GF algorithm in each generic kernel, median, average, and standard deviation are calculated in Figure 2.3 (d). The best GF kernel represented in the Tree structure is shown in Figs. 2, 3F on the scaled prostate dataset and with a setting hyperparameter.

## DISCUSSION

The proposed GF was deployed for the first time in the prostate/breast cancer detection dataset, and it demonstrated a significant performance improvement over the existing models in this domain. Experiments were carried out with a broad set of features from the conducted datasets. Table 6 gives a comparative description of several ML algorithms based on the same preprocessing conditions. The accuracy performance of the proposed GF model was superior compared to the six ML approaches in the prostate cancer dataset. Furthermore, we expanded this comparison with the suggested hybrid model by using the SVM classifier with several conventional kernels such as linear, polynomial, and RBF kernels.

The proposed model achieved 96.0% accuracy in the prostate cancer dataset, which is better than the ANN by 16% and the LR by 6%. The proposed GF beat the best-preset kernel by 8%, 20%, and 16% compared to linear, RBF, and polynomial kernels, respectively. The best kernel found for prostate cancer dataset (shown Fig. 2F) is:

['Plus_s', 'Multi_s', 'Multi_s', 'Plus_s', 'Minus_s', 'Minus_v', 'Minus_v', 'Plus_s', 'Plus_s', 'x', 'Minus_s', 'Plus_v', 'Plus_v', 'Plus_v', 'x', 'Plus_s', 'Minus_s', 'Minus_v', 'Minus_v', 'Minus_v', 'x', 'x', 'x', 'x', 'x', 'x', 'x']

The folding indicis of the best GF chromosome found for the produced kernel were:

['1.2', '3.4', '5.6', '7.8', '9.10', '0.5', '0.6', '11.12', '13.14', '0.9', '15.16', '17.18', '19.20', '21.22', '0.14', '23.24', '25.26', '0.17', '0.18', '0.19', '0.20', '0.21', '0.22', '0.23', '0.24', '0.25', '0.26']

In the same side, the best kernel string found for breast cancer dataset was shown in Fig. 3F:

['Plus_s', 'Plus_s', 'x', 'Plus_s', 'Multi_s', 'Multi_s', 'Minus_v', 'Plus_s', 'Plus_s', 'x', 'Minus_v', 'x', 'x', 'x', 'Minus_v']

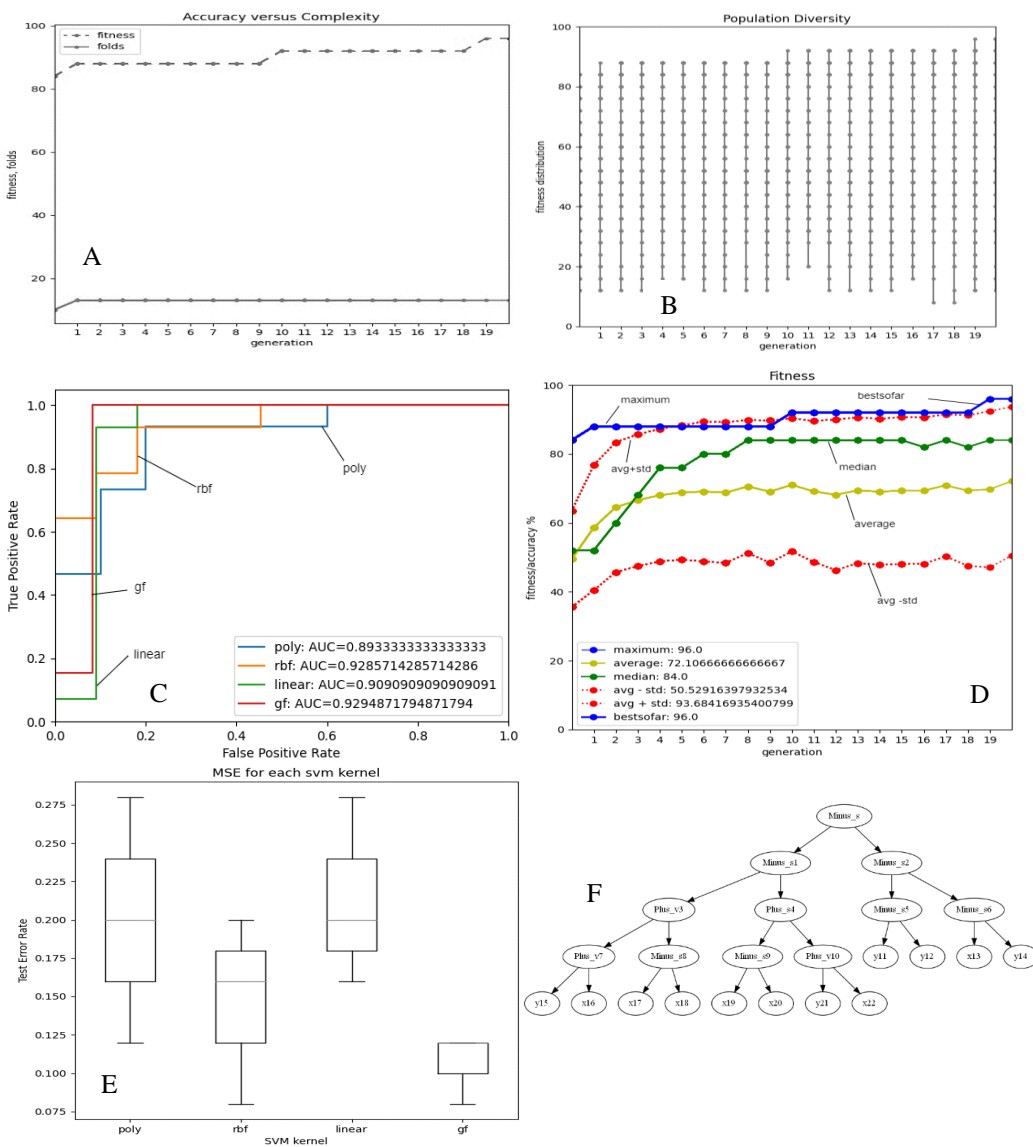

**Figure 2  GF algorithm results for Prostate cancer dataset.** (A) Accuracy *versus* complexity (B) Population diversity (C) The area under the curve (AUC) (D) Accuracy values (E) MSE values for each SVM kernel (F) GF kernel function.

The folding indices of the best GF chromosome found for the breast cancer dataset were: ['1.2', '3.4', '0.2', '5.6', '7.8', '9.10', '0.6', '11.12', '13.14', '0.9', '0.10', '0.11', '0.12', '0.13', '0.14']

As seen in Fig. 4, the proposed GF model performed the best for prostate cancer classification. After using five folds, the absolute average accuracy on the prostate

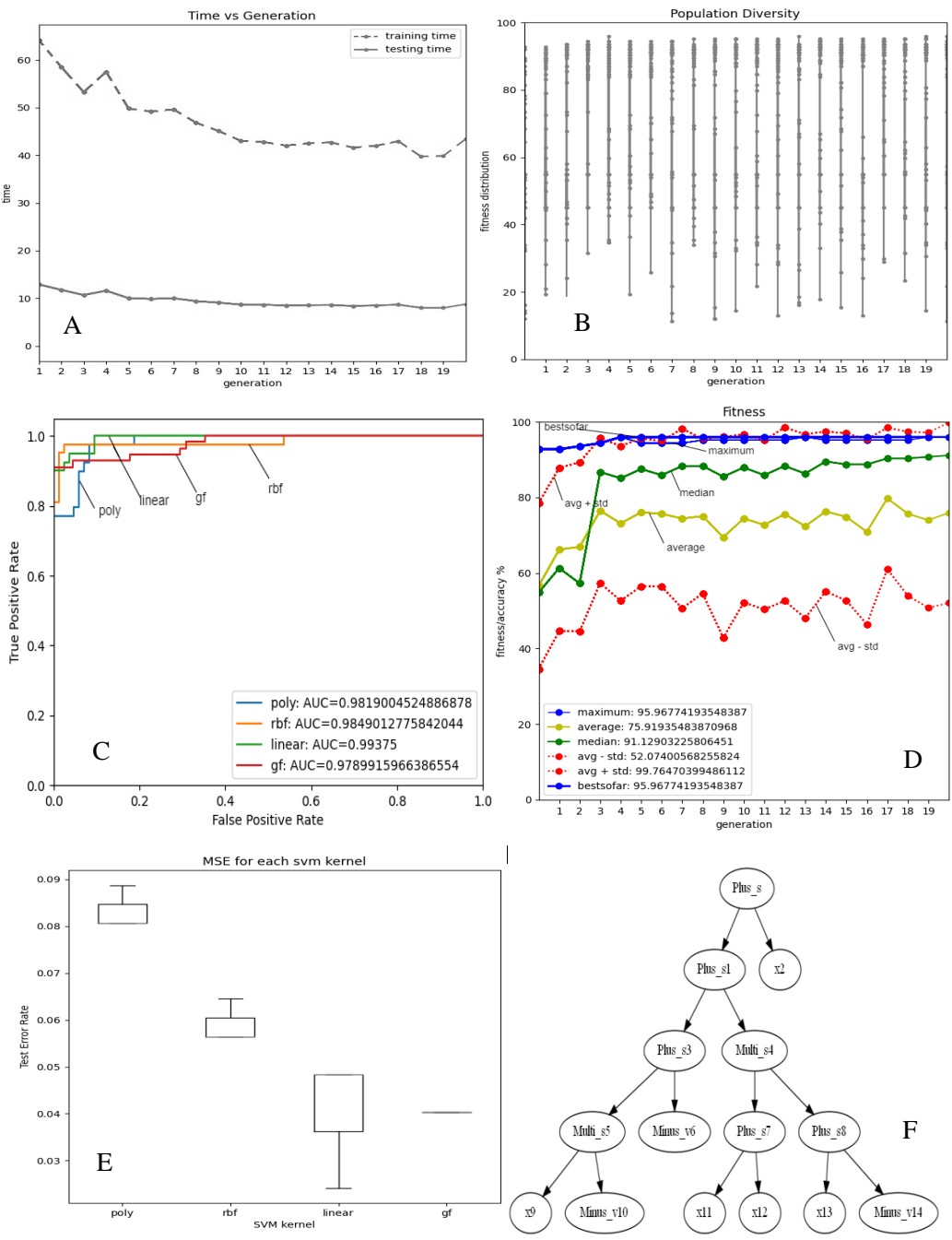

**Figure 3** **GF algorithm results for Breast cancer dataset.** (A) Accuracy *versus* complexity (B) Population diversity (C) The area under the curve (AUC) (D) Accuracy values (E) MSE values for each SVM kernel (F) GF kernel function.

cancer dataset was 96.0%. Additionally, when evaluated on a prostate cancer dataset, dimensionality reduction had an enhancing impact.

Consequently, the RF model achieved an accuracy of 96.0%, more significant than the proposed model by 0.09%, which was the best result. The accuracy comparison for

**Table 6** Comparisons between different kernel functions and AI models for the prostate cancer dataset.

| Model | Reference | Eliminated features | Accuracy (%) |
|---|---|---|---|
| Adaboost | *Cijov (2021)* | None | 75.8% |
| KNeighbors | *Gomes (2021)* | ['fractal_dimension', 'texture', 'perimeter'] | 71.0% |
| SVM (Linear) | GFLibPy | None | 88.0% |
| SVM (RBF) | GFLibPy | None | 76.0% |
| SVM (Polynomial) | GFLibPy | None | 80.0% |
| Random Forest (RF) | *Gomes (2021)* | ['fractal_dimension', 'texture', 'perimeter'] | 86.0% |
| Decision Tree (DT) | *Gomes (2021)* | ['fractal_dimension', 'texture', 'perimeter'] | 86.9% |
| Logistic Regression (LR) | *Ribeiro (2021)* | ['Area'] | 90.0% |
| Proposed GF | Our article | None | **96.0%** |
| Artificial Neural Network (ANN) | *Şenol (2021)* | None | 80.0% |

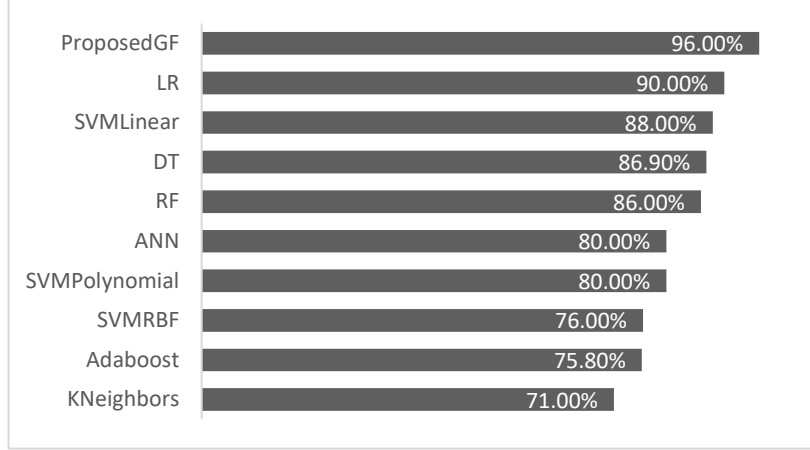

**Figure 4** Accuracies comparisons for prostate cancer dataset.

breast cancer is shown in Fig. 5, where all features are included. The proposed model was validated using just five folds, while the RF model-averaged accuracies ranged from 0-fold to 20-folds.

Although our system does not get the highest accuracy results in the breast cancer dataset, it is a close second. To emphasize the relevance of our findings, we compare the *P*-values of the hypothesis tests for both predefined SVM kernels (Figs. 6A and 6B) and ML models (Figs. 6C and 6D) to the reported *P*-value results for the proposed models in Figs. 4 and 5. Except for the prostate cancer dataset, all results are significant compared to a standard $P = 0.05$ criterion. Except for RBF, the SVM kernels in both datasets have an insignificant structure to GF. Furthermore, in both SVM kernels and ML models, the dataset size played the most critical role in GF's ability to forecast all negative samples and a representative sample for each positive sample. Figure 6 displays the *P*-value results for

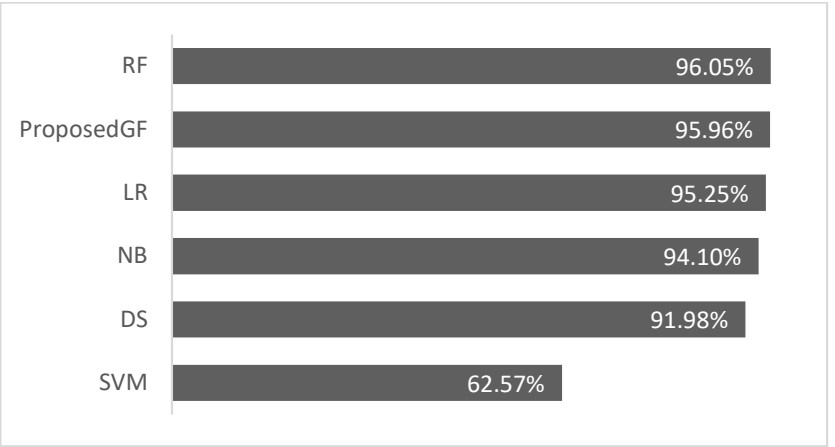

**Figure 5** Accuracies comparisons for breast cancer dataset.

both the prostate and cancer datasets compared to the predefined kernel-SVM and ML models.

## CONCLUSIONS

This article demonstrated that using a GF algorithm to classify patients with prostate cancer provides better accuracy than KNN, SVM, DT, and LR models. The GF algorithm achieved an average accuracy of 96% without eliminating any features from the dataset. This enables healthcare professionals to classify prostate cancers more precisely and provide more targeted therapies. Further improvements can be introduced to the model's accuracy based on our results. Using multidimensional data whilst choosing a range of feature selection/classification algorithms can be a promising tool for early-onset prostate cancer classification. We plan to work on the concept further and apply it to other types of cancer.

### Funding
The authors received no funding for this work.

### Competing Interests
The authors declare there are no competing interests.

### Author Contributions
- Mohammad A. Mezher conceived and designed the experiments, performed the experiments, performed the computation work, prepared figures and/or tables, authored or reviewed drafts of the article, and approved the final draft.
- Almothana Altamimi conceived and designed the experiments, performed the experiments, analyzed the data, prepared figures and/or tables, authored or reviewed drafts of the article, and approved the final draft.

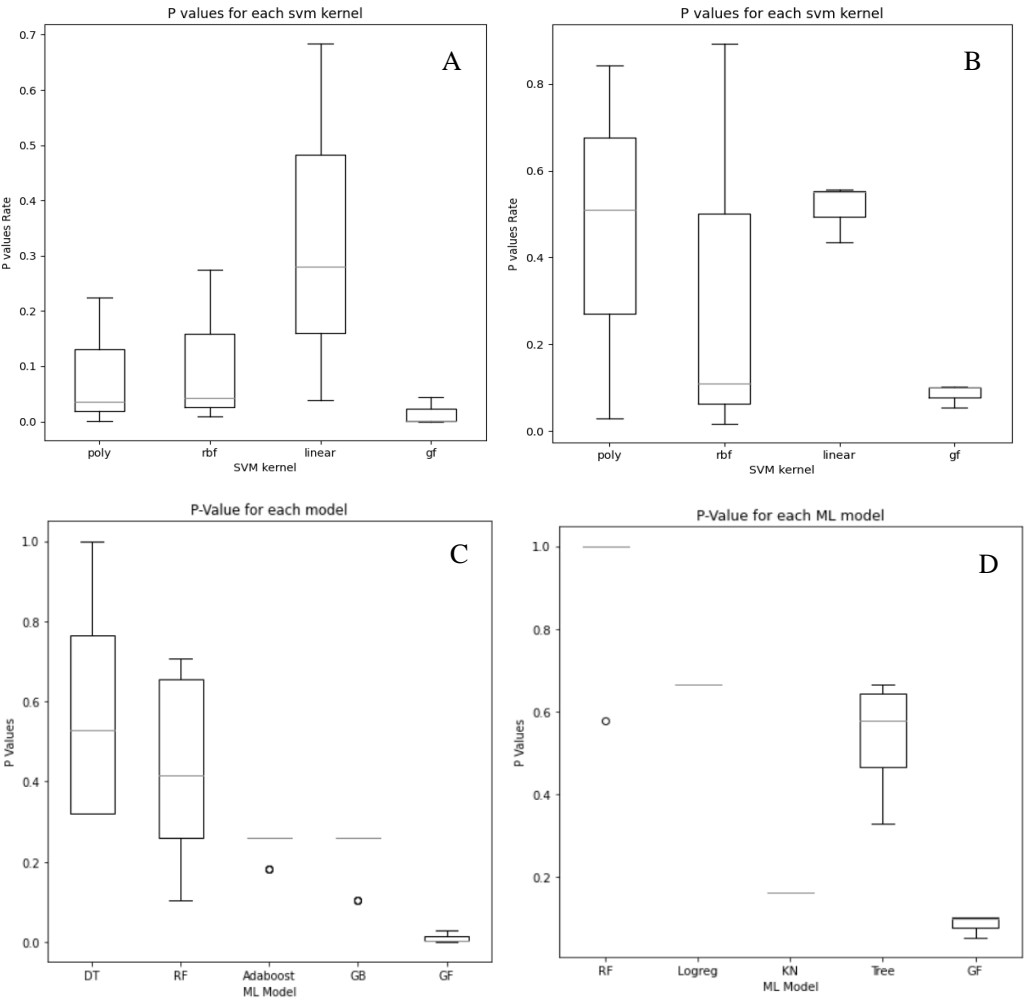

**Figure 6** *P*-values comparisons for prostate cancer (A, B) and breast cancer (C, D) dataset. (A) GF breast cancer *vs. Mezher (2022)* (B) GF prostate cancer *vs. Mezher (2022)* (C) GF breast cancer *vs. Anjelito & Mezher (2022)* (D) GF prostate cancer *vs. Smogomes & Mezher (2022)*.

- Ruhaifa Altamimi analyzed the data, prepared figures and/or tables, authored or reviewed drafts of the article, and approved the final draft.

## Data Availability

The data is available at Kaggle: https://www.kaggle.com/sajidsaifi/prostate-cancer.

The GF Code is available at GitHub: https://github.com/mohabedalgani/GFlibpy.

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
