# Peer review of "An enhanced Genetic Folding algorithm for prostate and breast cancer detection"

_PeerJ Computer Science, doi:10.7717/peerj-cs.1015_

## Round 0.1 · original submission · Major Revisions

Two reviewers studied your article and indicated serious issues about its content. At this stage, your manuscript cannot be accepted for publication.

Please address the issues raised by the reviewers and we will reassess your manuscript.

Reviewer 1 ·

Basic reporting

The manuscript has presented a clear meaning and has utilised professional English at most of the contents. However, there are concerns regarding the methodology and dataset:
1. how is the Genetic Folding algorithm developed in terms of its novelty, targeted issues in the dataset?
2. The dataset appears to be small rather than comprehensive to reflect the effectiveness of the algorithm. One another better dataset example can be:
"Using deep learning to enhance cancer diagnosis and classification"
3. The figures could not exclusively identify the details for the paper.
4. The algorithms could not be translated properly with its current form.
5. Please consider to make the reference clickable. For some reasons, I could not find the reference for Table 5.

Experimental design

Overall, it appears to be a work representing the quality of student project report, which has sufficiently reflected the terms and knowledge for building classifiers for particular dataset.
However, it fails to reflect the studied research questions, particularly it is difficult to understand the challenge to build a qualified and SOTA classifier for the given dataset. Meanwhile, it lacks sufficient details concerning the novelty and completeness of the utilised method. It appears to be an application work.

Validity of the findings

It is not strong enough to support the conclusion as SOTA performance for the given dataset.

Reviewer 2 ·

Basic reporting

In this MS the authors apply an evolutionary Genetic Folding (GF)
algorithm to the binary classification of prostate cancer malignancy from clinical tumour features.
The development and improvement of tumour stratification algorithms is an active area of research where advances might strongly benefit cancer diagnosis and treatment.

The authors provide a clear and concise overview of prostate cancer diagnosis and grading approaches, both clinical and computational.

The authors then propose and implement a SVM with GF-kernel based prostate cancer classifier (GF-SVM), arguing that GF has been shown to outperform other evolutionary algorithms.

A few passages should be edited using more appropriate terminology (e.g. line 174 should read "Instances were classified as...").

There are a few typos I would recommend addressing before publication (e.g. repetition at lines 78-78, line 134 should reference Mezher et. al 2010).

Experimental design

no comment

Validity of the findings

The authors state (lines 205,206) that GF-SVM "demonstrated a significant performance improvement over the existing models in this domain." However, the authors have failed to perform any sort of statistical analysis to demonstrate that their method is significantly better, or even different from, the established methods (e.g. see [1]). I would recommend the authors included a pairwise analysis of all alternative models in Fig. 2C versus GF-SVM (e.g. via a binomial McNemar test [2] over at least 10 holdout shuffle replicates). Without this I do not believe this manuscript meets the "statistically sound" criterion and I cannot recommend it for publication.

[1] Nicholls, Anthony. "Confidence limits, error bars and method comparison in molecular modeling. Part 1: the calculation of confidence intervals." Journal of computer-aided molecular design 28.9 (2014): 887-918.
[2] McNemar, Quinn, 1947. "Note on the sampling error of the difference between correlated proportions or percentages". Psychometrika. 12 (2): 153–157

Additional comments

The authors collect and made available a tabular dataset of 8 clinical features across 100 prostate cancer patients, as well as a repository for the Genetic Folding Open source library. Unfortunately, they do not provide the reader with any way of rapidly and conveniently reproducing their models and results (e.g. via a colab or jupyter notebook). I believe this would greatly help their model's chances of being re-used by others in the community.

---

## Round 0.2 · Minor Revisions

The article has improved but it cannot be accepted for publication yet.
Please address the requests by the reviewers, adding a part about technical contributions and analysis made with statistical tests.

Reviewer 1 ·

Basic reporting

Thanks for the revision efforts. It is found that more details have been added, however, it lacks a section of describing the distinguishable technical contributions.
Moreover, the reference is very limited. Please consider including more up-to-date reference, such as:
Weakly supervised prostate tma classification via graph convolutional networks;
Integrating genomic data and pathological images to effectively predict breast cancer clinical outcome;
Supervised machine learning model for high dimensional gene data in colon cancer detection;
A survey on machine learning approaches in gene expression classification in modelling computational diagnostic system for complex diseases;
OncoNetExplainer: explainable predictions of cancer types based on gene expression data;
A Novel Statistical Feature Selection Measure for Decision Tree Models on Microarray Cancer Detection;
Selecting features subsets based on support vector machine-recursive features elimination and one dimensional-Naïve Bayes classifier using support vector machines for classification of prostate and breast cancer.

Experimental design

Please refer to the mentioned reference regarding the experimental design. The p-value will be desired to evaluate the effectiveness of the model.

Validity of the findings

It could be validated if the source code and datasets are publicly available.

Reviewer 2 ·

Basic reporting

The authors have expanded their model performance analysis by including a pairwise model accuracy comparison among a set of alternative SVC kernels and their proposed GF kernel in Fig. 2C,E. This analysis shows that GF outperforms all other evaluated kernels on these classification problems. I believe the manuscript now satisfies this journal's criterion for publication. However, I would personally still recommend the authors included some sort of pairwise statistical test in this analysis (e.g. Mann-Whitney, McNemar) to robustly assess the significance of the observed improvement in performance.

Experimental design

no comment

Validity of the findings

no comment

Additional comments

no comment

---

## Round 0.3 · accepted · Accept

The authors addressed correctly the points raised by the reviewers, so I recommend this article to be accepted for publication. The tests and machine learning applications have been carried on in a sound way.